# PET-CT in Clinical Adult Oncology: III. Gastrointestinal Malignancies

**DOI:** 10.3390/cancers14112668

**Published:** 2022-05-27

**Authors:** Bhasker R. Koppula, Gabriel C. Fine, Ahmed Ebada Salem, Matthew F. Covington, Richard H. Wiggins, John M. Hoffman, Kathryn A. Morton

**Affiliations:** 1Department of Radiology and Imaging Sciences, University of Utah, Salt Lake City, UT 84132, USA; bhasker.koppula@hsc.utah.edu (B.R.K.); gabriel.fine@hsc.utah.edu (G.C.F.); ahmed.salem@utah.edu (A.E.S.); matthew.covington@hsc.utah.edu (M.F.C.); richard.wiggins@hsc.utah.edu (R.H.W.); john.hoffman@hci.utah.edu (J.M.H.); 2Department of Radio Diagnosis and Intervention, Faculty of Medicine, Alexandria University, Alexandria 21526, Egypt; 3Summit Physician Specialists, Intermountain Healthcare Hospitals, Murray, UT 84123, USA

**Keywords:** PET, FDG, esophageal cancer, gastric cancer, hepatobiliary carcinoma, small-bowel carcinoma, colorectal cancer pancreatic cancer, adrenal carcinoma, anal carcinoma

## Abstract

**Simple Summary:**

Positron emission tomography (PET), typically combined with computed tomography (CT), has become a critical advanced imaging technique in oncology. With PET-CT, a radioactive molecule (radiotracer) is injected in the bloodstream and localizes to sites of tumor because of specific cellular features of the tumor that accumulate the targeting radiotracer. The CT scan, performed at the same time, provides information to facilitate the characterization of radioactivity from deep or dense structures, and to provide detailed anatomic information. PET-CT has a variety of applications in oncology, including staging, therapeutic response assessment, restaging and surveillance. This series of six review articles provides an overview of the value, applications, and imaging interpretive strategies of PET-CT in the more common adult malignancies. The third report in this series provides a review of PET-CT imaging in gastrointestinal malignancies.

**Abstract:**

PET-CT is an advanced imaging modality with many oncologic applications, including staging, assessment of response to therapy, restaging and longitudinal surveillance for recurrence. The goal of this series of six review articles is to provide practical information to providers and imaging professionals regarding the best use of PET-CT for specific oncologic indications, and the potential pitfalls and nuances that characterize these applications. In the third of these review articles, key tumor-specific clinical information and representative PET-CT images are provided to outline the role that PET-CT plays in the management of patients with gastrointestinal malignancies. The focus is on the use of ^18^F fluorodeoxyglucose (FDG), rather than on research radiopharmaceuticals under development. Many different types of gastrointestinal tumors exist, both pediatric and adult. A discussion of the role of FDG PET-CT for all of these is beyond the scope of this review. Rather, this article focuses on the most common adult gastrointestinal malignancies that may be encountered in clinical practice. The information provided here will provide information outlining the appropriate role of PET-CT in the clinical management of patients with gastrointestinal malignancies for healthcare professionals caring for adult cancer patients. It also addresses the nuances and provides interpretive guidance related to PET-CT for imaging providers, including radiologists, nuclear medicine physicians and their trainees.

## 1. Introduction

PET-CT is an invaluable advanced diagnostic imaging modality in oncology with a variety of applications, including initial staging of cancer, assessment of response to therapy, restaging and longitudinal surveillance for recurrence. The goal of this six-part review is to provide practical information for referring providers, radiologists, nuclear medicine practitioners and their trainees regarding the best use of PET-CT for specific oncologic indications, potential pitfalls and nuances, and guidance in interpreting these scans accurately. The third in this series of six review articles focuses specifically on gastrointestinal (GI) tumors, including esophageal, stomach, small bowel, colon, anal, hepatic, pancreatic, adrenal and gastrointestinal stromal (GIST) tumors. Gastroenteropancreatic neuroendocrine tumors (GEP NET), and adrenal and extra-adrenal pheochromocytomas/paragangliomas are discussed separately in the sixth review article in this series. The emphasis is on the use of ^18^F fluorodeoxyglucose (FDG), which is used almost exclusively for GI tumor imaging, rather that research tracers under development. It is acknowledged that there is a large body of compelling literature that relates FDG PET-CT characteristics to prognostic factors for specific gastrointestinal cancers. This is not the focus of this review, because the management of these patients is typically dictated by the histology, patient condition and extent of disease. Because PET-MR scanners are in limited use, the focus here is on PET-CT because if its widespread availability. However, basic principles described are also applicable to PET-MR.

### Esophageal and GE Junction Cancer

Esophageal cancer is the sixth leading cause of cancer deaths worldwide, with an incidence that is increasing. Squamous cell carcinoma is the most common type, accounting for 90% of all esophageal cancers, typically involving the upper (cervical and upper thoracic) and mid-esophagus, with risk factors of smoking, chewing tobacco, excessive consumption of alcohol and a diet low in fruit and vegetables [1]. Esophageal adenocarcinoma typically occurs in the distal esophagus or gastroesophageal (GE) junction, often extending into the cardia of the stomach, with risk factors of chronic gastroesophageal reflux disease (GERD) with Barrett’s metaplasia. Curative treatment of esophageal cancer typically relies upon neoadjuvant chemoradiation, followed by surgical resection. For patients who are poor surgical candidates, curative treatment by definitive chemoradiation may be an alternative. Very early-stage disease may be managed by endoscopic minimally invasive techniques, such as mucosal resection [2]. Metastatic disease is typically managed by chemotherapy with radiation reserved for local control and complications. Accuracy in initial staging and assessment of response to initial chemoradiation is critical in the management of patients with esophageal cancer. PET-CT is an important modality in both applications.

There are several pitfalls and both physiological and benign patterns FDG uptake that can complicate interpretation of PET-CT for esophageal cancer [3,4,5]. The normal esophagus may have mild diffuse uptake, which is often most pronounced at the GE junction. Esophagitis, esophageal inflammatory erosions, uptake within hiatal hernias, in regions of Barrett’s metaplasia, and in post-treatment fistulas may also result in false-positive scans. Benign esophageal leiomyomas may show uptake of FDG [6]. In our experience, the tissue at the proximal and distal end of esophageal stents may show increased uptake due to pressure effects. It is advisable to wait at least 10–12 weeks following radiation therapy to obtain a post-treatment PET-CT to avoid uptake due to radiation-induced inflammatory changes in the esophagus (Figure 1 and Figure 2). Adjacent tissue that may be within or adjacent to the radiation field, such as lung and liver, may also show hypermetabolism from radiation injury that can be confused with regional metastatic disease (Figure 3) [7]. Prediction of early metabolic response by an early (interim) FDG PET-CT may avoid false positives that occur post-treatment [8].

Many false-negative FDG PET-CT scans are also seen with esophageal cancer, due to superficial or small tumors and highly mucinous neoplasms. Depth of penetration and length of the tumor cannot be accurately assessed by PET or CT. For these determinations, esophageal endoscopic ultrasound (EUS) is preferable. In addition, for initial staging of esophageal cancer, both CT and FDG PET-CT are limited in detection of regional nodal metastases in the immediate vicinity of the primary tumor, with a range in sensitivity and specificity for CT and FDG PET-CT of 41–60% and 77–89%, and a range in sensitivity and specificity for CT or FDG PET-CT of 43–70% and 76–95% [9]. For detection of distant metastases, FDG PET-CT offers a clear advantage over CT [9].

Patterns of initial nodal spread [10] depend upon the location of the primary tumor. Cervical esophageal cancers typically spread first to paraoesophageal cervical or upper thoracic nodes, as well as low anterior cervical and supraclavicular nodes (Figure 4). Mid-esophageal tumors spread first to paraoesophageal nodes or subcarinal nodes near the primary site. GE junction tumors spread to paraoesophageal nodes adjacent to the primary tumor, to the gastric cardia and to gastrohepatic nodes (Figure 5). More invasive or recurrent tumors may be trans-spatial and infiltrative, and, for mid-esophageal lesions, may involve the heart and aorta (Figure 6). Distant metastases typically first involve the liver and lungs.

The best use of FDG PET-CT for esophageal and GE junction tumors is currently reflected by NCCN recommendations for three indications: for initial diagnostic staging (for distant disease); for treatment response assessment preoperatively following neoadjuvant chemoradiation, and for treatment assessment following definitive chemoradiation [11]. However, there is uncertainty as to whether post-neoadjuvant chemoradiation PET-CT parameters can adequately identify patients with pathologic complete response [12]. The value of PET-CT for surveillance or assessment of recurrence has not been as well established, although it may be helpful in individual cases. However, once recurrence is identified, PET-CT has an advantage over other modalities in again identifying distant metastatic disease.

## 2. Gastric Carcinoma

Gastric cancer is the second most common cause of cancer death and the fourth most common cancer in the world, with a high prevalence in Eastern Asia. Rates in the US are significantly lower than those worldwide [13,14]. There are many factors that have been linked to an increased risk of gastric cancer, among which are atrophic gastritis, chronic H. pylori infection and dietary factors such as high salt and preservatives. Ninety percent of gastric cancers are adenocarcinoma, which is the focus of this discussion. Cure can only be achieved by resection and lymph node dissection, so accurate staging is important. However, whether PET-CT can provide that is somewhat debatable.

There are two major histological types of gastric carcinoma, diffuse (non-cohesive, undifferentiated) and intestinal (cohesive, differentiated) according to the Lauren and Nakamura classification systems [15,16]. There has also been the suggestion that the differentiated phenotype can alternatively be divided into gastric and intestinal subtypes, related to the cell of origin and the presence of mucin (gastric subtype) [17]. The metabolic profile of gastric carcinoma is highly variable by FDG PET-CT, and may reflect molecular and histologic growth-type differences in gastric carcinoma [18]. Whether FDG PET-CT is useful in staging non-GE junction gastric cancer is controversial. There are many benign causes of increased uptake by the stomach on FDG PET-CT, including normal physiological uptake, gastritis, ulcers and unknown factors (Figure 7). If diffuse or focal metabolic activity is noticed on an FDG PET-CT scan for unrelated reasons, clinical evaluation is recommended and endoscopy if there are clinical gastric symptoms. For detecting the primary tumor at an early stage, FDG PET-CT sensitivity is generally poor, reported to range from 26 to 63%, and is worse for mucin-producing tumors [19,20,21]. Although some gastric cancers can be markedly metabolically active, many others display metabolic activity that is only mild and within the range of normal gastric activity (Figure 8 and Figure 9). Gastrinomas may mimic gastric cancer, with hypertrophy of the fundal parietal cells, often associated with a hypersecretory state that can mimic bowel obstruction (Figure 10). Linitus plastica, with diffuse neoplastic infiltration of the stomach, typically by adenocarcinoma, may appear and a diffuse thickening that is only mild-moderately hypermetabolic (Figure 11). For initial T staging, EUS is considered the best modality, with a sensitivity of 82%. The sensitivity and specificity of endoscopic ultrasound (EUS) and multidetector CT (MDCT) for detecting regional nodes is 91% and 49% for EUS, and 77% and 63% for MDCT [22]. Even in staging for regional or distant spread of disease, there is debate as to the value of PET-CT. It has been reported that PET-CT identifies positive nodes not detected by CT in only 5%, although it predicts poorer prognosis when positive. The specificity is reported to be higher with PET than CT [23,24]. Others have proposed that FDG PET-CT when performed with a contrast-enhanced, high-quality CT scan offers the best imaging approach, with advantages of both PET and CT [25].

The curative-intent post-surgical recurrence rate for gastric carcinoma is in the range of 22–60%, for which PET-CT has been reported to have a sensitivity of 91.2% and a specificity of 61.5% [26]. Early response to treatment may also be predicted by an “interim” FDG PET-CT at two weeks post-initiation of chemotherapy, with an SUV reduction of >35% resulting in a positive predictive value (PPV) of 86% and a negative predictive value (NPV) of 77% [27]. Despite the lack of strong evidence, the NCCN guidelines for the use of FDG PET-CT in gastric carcinoma are quite permissive, supporting the FDG PET-CT for four indications: (1) for initial staging if there is no evidence of M1 disease and if clinically indicated; (2) for preoperative assessment of response to neoadjuvant chemotherapy or chemotherapy; (3) Restaging following adjuvant chemotherapy or chemoradiation in unresectable patients or non-surgical candidates, (4) surveillance following neoadjuvant +/− adjuvant therapy in stage I-III disease [28]. In summary, the value of FDG PET-CT in the initial staging of gastric cancer is controversial. Nonetheless, there is significant opportunity to use the modality in the management of gastric carcinoma based on perceived medical need in specific patients. An awareness of the limitations of PET in this disease are important in the proper management of patients with gastric carcinoma.

## 3. Gastrointestinal Stromal Tumor (GIST)

Gastrointestinal stromal tumors (GISTs) are rare solid tumors that occur anywhere in the GI tract. Most often discovered incidentally, they can either display benign characteristics or undergo transformation to acquire malignant phenotypic features [29]. These tumors do not respond to radiation or chemotherapy. Treatment is typically surgical, with first or second-generation tyrosine kinase receptor inhibitor (TKI) therapy reserved for non-surgical cases or metastatic spread.

Significant emphasis is placed on determining whether GISTs are benign or malignant. Patients with high-risk tumors are often place on TKI’s even after surgical resection to prevent recurrence. However, the distinction between benign and malignant GISTs is fraught with difficulty in that all GISTs are potentially malignant (particularly those of extra-gastric origin), and the classification of these tumors has undergone frequent change. Factors that are associated with malignant GIST phenotypes are large size of the tumor (>5 cm), the presence of necrosis, heterogeneous enhancement and metastases. FDG PET-CT offers an advantage over other modalities in metastatic staging, in an assessment of necrosis and in detection of liver metastases [30]. GISTs are typically strongly hypermetabolic on FDG PET-CT. However, GISTs with low uptake of FDG have been reported and lack of significant uptake should not exclude the diagnosis of GIST [31].

There are challenges and competing strategies in assessing response to treatment of GISTs, the most significant of which are conventional RECIST and Choi criteria (Table 1). The Choi criteria proposes that a decrease in attenuation of the tumor is an important predictor of tumor response and is generally viewed as a more robust indicator of response (30).

The use of FDG PET-CT in evaluating response to TKI inhibitor therapy in GISTs offers several advantages in that decreased metabolic activity, and indicator of response, can be seen significantly earlier than a decrease in size of the tumors [31], (Figure 12 and Figure 13). FDG PET has also been proposed as an early prognostic indicator response to Imatinib Mesylate. For example, at one month post-treatment, a reduction in SUVmax of ≥40% (*p* = 0.002) or an SUVmax of ≤3.4 (*p* = 0.00002) is highly indicative of response [32]. However, complete metabolic response does not indicate a complete pathologic response, but rather disease control. It should be noted that ^18^F fluciclovine and both ^18^F and ^68^Ga PSMA-binding PET agents developed and approved for prostate cancer imaging show intense uptake in GIST tumors. However, the utility of this agent in GIST tumors, or advantages over FDG.

## 4. Hepatocellular Carcinoma

Hepatocellular carcinoma (HCC) is the most common primary malignancy of the liver and the most common visceral tumor worldwide. It is also the second most common malignancy of the liver in children, after hepatoblastoma. In adults, HCC often arises as a consequence of chronic liver disease, such as alcoholic cirrhosis and chronic hepatitis B or C. With increasing frequency of obesity and diabetes, non-alcoholic steatohepatitis (NASH) is becoming an increasingly more important cause of HCC.

HCC can present as a large single mass, often with necrosis, fat elements and variable calcification. It can also present as multinodular/multifocal disease with or without central necrosis. Finally, it can present as diffusely infiltrating disease, which is difficult to distinguish from underlying cirrhosis. Imaging strategies typically rely upon contrast-enhanced CT (CE CT), which typically demonstrates an enhancing mass with portal vein invasion, late arterial enhancement with rapid washout, or arterioportal shunt with wedge-shaped perfusion defects associated with focal steatosis or focal fatty sparing. Fibrolamellar HCC may show a central scar, similar to that of focal nodular hyperplasia. MR imaging is often employed for characterization, and shows T1 + Gd arterial enhancement with rapid washout, often with a persistent rim of enhancement. Diffusion weighted imaging (DWI) MR showing a high intratumoral signal is also suggestive of HCC. Although highly specific, the typical enhancement pattern seen with CE CT or MR has a sensitivity of only 60% [33].

Unfortunately, the value of FDG PET in HCC is limited. Moderately or well-differentiated HCC is typically isometabolic to the liver (Figure 14). However, poorly differentiated HCC in the liver, and its metastases, tend to be hypermetabolic relative to the liver. PET may provide useful information in identifying regions of more poorly differentiated tumor within a known HCC or in excluding distant metastases in patients with poorly differentiated HCC who are considered for liver transplantation (Figure 15). Of note, adenomas and cavernous hemangiomas of the liver are also typically isometabolic to the normal liver parenchyma on FDG PET (Figure 16 and Figure 17). Other PET tracers, such as ^18^F choline, have been shown to have improved performance over FDG PET for well-differentiated HCC [34,35]. Several recent reports support that PSMA PET ligands may be of value in imaging HCC, including well-differentiated forms [36,37,38,39]. PSMA PET-CT, typically used in prostate cancer, is potential marker of neovascularity in the HCC and a potential future theragnostic target for HCC. However, FNH and hepatic hemangiomas are can also be positive on PSMA PET, which could limit specificity [40].

Treatment for limited-stage HCC (<5 cm, no metastases) includes resection and liver transplantation. Advanced disease is managed by systemic therapy, depending on degree of hepatic reserve. The systemic therapy typically involves the use of multiple agents, including TKIs, as well as targeting PD-1, check-point and VEGF inhibitors in various combinations. Disease control may be attempted with liver-direction interventions, including bland embolization, Y-90 microsphere radioembolization, or transarterial chemoembolization (TACE), either with curative or palliative intent [41].

## 5. Cholangiocarcinoma

Cholangiocarcinoma (CC) arises from the bile ducts [42]. CC accounts for 10–15% of hepatic malignancies. The disease has a high prevalence in Asian countries, thought due to endemic parasitic diseases. Chronic inflammatory conditions of the biliary tract, including chronic cholecystitis, primary sclerosing cholangitis and ulcerative colitis may play a role although the causes of most cases of CC are unknown.

Adenocarcinoma accounts for 90% of cases of CC, with 10% attributable to squamous cell carcinoma. There is a complex classification system for CC. Anatomically, tumors can be classified as either intrahepatic, extrahepatic (hilar, Klatskin tumors) or distal extrahepatic (common bile duct) [43,44]. Further classification systems of CC have also been developed that pertain to macroscopic growth pattern, microscopic features and cell of origin.

The prognosis with CCC is generally poor. The only cure for CC is surgical resection. Unfortunately, typically CCC presents late with advanced, inoperable disease [45]. Management is then primarily palliative, and consists of primarily of biliary drainage, with photodynamic therapy, radiation therapy, chemotherapy and molecular targeted therapies employed in some cases [46].

The utility of FDG PET-CT in CC is variable. Intrahepatic CC is typically intensely hypermetabolic and may present as large, ring-shaped masses (Figure 18). Hilar CC is typically lower in activity, occasionally with a branching pattern (Figure 19). There are false positives in imaging CC by FDG PET-CT. Biliary obstruction may lead to focal bacterial cholangitis due to obstructed biliary radicles, which may appear hypermetabolic on FDG PET-CT (Figure 20). Hypermetabolic host-reactive change around biliary stents may also be confused with tumor infiltration.

Although FDG PET-CT offers no advantage in the diagnosis of CC, it may be of benefit in staging and detection of distant metastases [47,48]. A recent comparison of guidelines for use of FDG PET-CT by the National Comprehensive Cancer Network (NCCN), The European Society for Medical Oncology (ESMO), The British Society of Gastroenterology (BSG) and the International Liver Cancer Association (ILCA) reported that all organizations supported the use of MRCP and CECT for management of cholangiocarcinoma. Only the NCCN supported the use of FDG PET-CT as a staging modality [41]. Despite this, FDG PET-CT has been shown to affect management of some patients with CC [49,50]. For example, FDG PET-CT may help direct targeted biopsy, or help direct treatment strategies between curative and palliative. As such, FDG PET-CT may be of value in selected patients.

## 6. Gallbladder Carcinoma

Gallbladder carcinoma is an epithelial mucosal malignancy that is similar to cholangiocarcinoma in histology, with up to 97% being adenocarcinomas [51]. It is the fifth most common GI tract malignancy. Risk factors include chronic inflammatory conditions of the biliary tree such as chronic cholecystitis and gallstones, as well as primary sclerosing cholangitis and some gallbladder polyps. Genetic factors may also play a role [52].

The overall prognosis of gallbladder cancer is similar to that of cholangiocarcinoma, with a 5% five-year survival. However, survival may be improved with accurate staging and early-stage-appropriate therapy. Most cases of gallbladder cancer are found incidentally at the time of a routine cholecystectomy, which occurs in up to 2–3% of cholecystectomies [53,54]. Surgical treatment depends on stage and may range from cholecystectomy for very early-stage carcinoma to a second stage surgery consisting of radical cholecystectomy with wedge resection of the surrounding liver and hepatoduodenal ligament lymphadenectomy [55,56]. Spread of tumor is initially to the liver with initial lymphatic spread to periportal and periaortic/pericaval lymph nodes at the level of the renal vessels [57]. Therapy for advanced or metastatic gallbladder cancer relies primarily upon conventional chemotherapy as well as clinical trials with check-point inhibitors, particularly in patients with DNA mismatch repair deficiency or microsatellite instability [58].

Typically, gallbladder cancer and its metastases are hypermetabolic on FDG PET-CT (Figure 21). There is a relative scarcity of data regarding the role of FDG PET-CT in gallbladder cancer. The NCCN does not mention FDG PET in the management of gallbladder cancer, but rather contrast-enhanced CT or MRI of the abdomen and pelvis, and a CT of the chest. However, a recent meta-analysis of available data of the performance of FDG PET-CT in gallbladder cancer reported a pooled sensitivity of 96% and specificity of 91% for local disease, and a pooled sensitivity and specificity for metastatic disease of 96% and 91%. For nodal disease, the pooled sensitivity was 75% and the specificity was 91% [59]. The data regarding nodal disease is significant in that regional nodes involved with gallbladder cancer tend to be small and present a challenge for CT. Although available literature supports that FDG PET-CT may play a significant role in the evaluation of gallbladder cancer, larger trials are warranted. It is important to know that false positives resulting in FDG uptake in the gallbladder exist, including acute and chronic cholecystitis, xanthogranulomatous cholecystitis and papillary hyperplasia (Figure 22 and Figure 23) [60,61]. Patients with stents in the common bile duct frequently show uptake in the wall of the gallbladder, likely secondary to inflammation and partial obstruction. Nonetheless, when anatomic imaging is equivocal, FDG PET-CT may be of value in the management of selected patients with gallbladder cancer [62].

## 7. Adrenal Cancer

Primary adrenal cancers arise from both the cortex and medulla. Medullary adrenal cancers consist predominantly of pheochromocytomas and neuroblastomas derived from chromaffin cell precursors [63]. These are addressed in the sixth article in this series of reviews. Adrenocortical carcinomas are rare tumors and are diverse in their characteristics. They can be hormonally non-functional or functional. Functional tumors may present with Cushing syndrome, virilization, or feminization. Adrenocortical carcinomas often initially present with advanced disease, with invasion into surrounding viscera (kidney, liver), IVC tumor thrombus, regional nodal disease and distant metastases. The most effective treatment is surgical resection. Advanced cases are typically treated with Mitotane, an adrenocytolytic agent or combined chemotherapeutic regimens [64].

Adrenocortical carcinomas and their metastases are typically strongly FDG avid. However, oncocytic adrenocortical carcinoma has been reported to be low in FDG uptake [65]. Therefore, low metabolic activity on FDG PET-CT does not preclude the possibility of adrenocortical carcinoma and metabolic signatures defined by FDG PET-CT that reliably distinguish adrenocortical carcinoma from other differential possibilities do not exist. Adrenocortical carcinomas are rare. When an adrenal mass or nodule is identified on FDG PET-CT and there is no evidence of a secretory abnormality, the statistical likelihood is far greater that the lesion is either a benign adrenal lesion or a metastasis from another source, rather than an adrenocortical carcinoma. A large size and other imaging characteristics defined by CT and MRI, such as heterogenous enhancement, necrosis and invasion into surrounding structures support that an adrenal mass is likely a malignancy (Figure 24). FDG PET-CT may play a role in the assessment in identifying a primary tumor when a probable adrenal metastasis is identified.

The more common challenge for FDG PET-CT is in establishing whether an adrenal nodule is more likely to be benign or malignant. The term “adrenal incidentaloma” is typically applied to adrenal nodules that are found in patients undergoing imaging for reasons other than cancer [66,67]. In the case of FDG PET-CT scans performed for oncologic purposes, the term “incidentaloma” is not clearly applicable. The likelihood that an adrenal nodule is cancerous in a patient with an underlying malignancy is 20%, and of those surgically removed because of high suspicion of malignancy, 30% are benign [66]. Therefore, there is a significant amount of uncertainty related to defining adrenal nodules as either benign or likely malignant, even in patients with known cancer.

CT and MRI criteria have been applied to identify benign adrenal incidentalomas. These are primarily focused on evaluating the lesion for the presence of fat, and indicator of a lipid-rich adenoma. The imaging method with the strongest evidence base for evaluating adrenal nodules >1 cm is the Hounsfield unit (HU) of the nodule on non-contrast-enhanced CT. A HU ≤ 10 is suggestive of a benign lipid-rich adenoma [66,67]. Although this indicator is of value in non-cancer patients, it has been shown to be less reliable in patients with malignancy, likely due to necrosis of adrenal tumors which then appear low in attenuation [66]. Necrosis usually is associated with heterogeneous attenuation and if the attenuation is low and uniform, the CT assessment is likely more accurate. A HU of >10 on non-contrast CT has a sensitivity of 100% for detecting malignant adrenal lesions but the specificity is so low as to make this criterion indeterminate. Multi-time point contrast-washout CT has also been performed for adrenal nodule assessment, and most benign adenomas show a washout of >50% in 10 min (compared to enhancement on the portal venous phase of contrast) [68].

In patients undergoing FDG PET-CT, criteria that attempt to differentiate between benign and malignant lesions have included SUVmax of the adrenal lesion, and an SUVmax ratio between the adrenal lesion and the liver. Absolute SUVmax cut-off values proposed range widely, from approximately 2.5 to 4.6, without a clear consensus for the cut-off value or prospective independent validation of cut-off values in large series [69,70]. The use of absolute SUVmax in establishing whether a lesion is benign or malignant could also vary depending on differences between cameras, imaging technique and patient-specific factors. Tumor:liver ratios of SUVmax of >1.5 have been shown to have a sensitivity and specificity of 82% and a specificity of 96% in detecting malignancy and are more commonly used in evaluation of adrenal masses [71]. In a patient undergoing an FDG PET-CT, the appearance on CT must be considered in concert with the PET findings in evaluating the likelihood that a lesion is benign. According to commonly used PET criteria, if a lesion shows very low uptake of FDG, or uptake that is less than the liver, the lesion is likely benign, recognizing that some tumors (low grade lymphoma, renal cell carcinoma, well-differentiated neuroendocrine tumors, and oncocytic adrenocortical carcinoma) may be low in FDG uptake. Greater confidence is provided by low FDG uptake with concordant CT findings of uniformly low attenuation. If a lesion is first identified and is felt clinically unlikely to be due to malignancy, routine endocrine evaluation should probably be performed. In cases with a low suspicion of malignancy, determination of stability on follow-up imaging is acceptable. In lesion with more suspicious imaging or clinical features, an absence of additional systemic metastases, and no contraindications to biopsy, tissue sampling may be required (Figure 25). Of note, large bilateral adrenal metastases place a patient at risk of adrenal insufficiency/adrenal crisis and expeditious communication regarding this risk is crucial so that the appropriate endocrinology assessment can be performed.

There are many false-positive hypermetabolic adrenal masses due to benign causes, including adrenal hemorrhage, tuberculomas, some secreting benign adrenal adenomas, ganglioneuromas, and some pheochromocytomas [72]. Diffuse hypermetabolism in an adrenal gland without a nodule also presents some challenges to FDG PET-CT. Following surgical resection of an adrenal gland, it is not unusual to observed diffuse increase in size and in FDG uptake in the other adrenal gland [73], possibly due to compensatory hypertrophy. Bilaterally symmetrical diffuse enlargement and hypermetabolism of the adrenal glands may represent adrenal hypertrophy or may be due to stress-induced adrenal activation. Asymmetrical enlargement and hypermetabolism of a single adrenal gland could be caused by an adrenal aldosteronoma and further evaluation as per typical endocrinology protocols is recommended if the patient is hypertensive.

## 8. Pancreatic Carcinoma

Pancreatic cancer is the fourth leading cause of cancer deaths and is more common in men than women [74]. Seventy-five percent occur in the pancreatic head and body, often resulting in obstruction of the common bile ducts. Symptoms are nonspecific, although painless jaundice or upper abdominal pain boring through to the back is suggestive. However, pancreatic cancer is typically clinically silent until it is advanced and, even when more advanced, can masquerade as pancreatitis, which may coexist with pancreatic cancer because of ductal obstruction by the mass. CECT and MRI are typically the initial imaging modalities used. The diagnosis is typically established by EUS and biopsy.

Pancreatic neoplasms consist primarily of ductal adenocarcinoma (85%), with 3% representing acinar cell carcinoma. Benign or indeterminate cystic neoplasms such as serous or mucinous cystadenomas and intraductal papillary mucinous neoplasms (IPMNs) represent approximately 10% of pancreatic neoplasms, and pancreatic neuroendocrine tumors (PNET, islet cell tumors) account for 2%. Rare metastatic tumors to the pancreas, such as with melanoma, may also occur. PNET tumors are discussed in the sixth article in this series of reviews. The following discussion is focused primarily on pancreatic ductal and acinar cell adenocarcinoma.

The NCCN endorses the use of FDG PET-CT in its guidelines for initial staging of high-risk patients with pancreatic cancer [75]. The sensitivity of detecting metastatic disease for PET-CT, standard CT, and combined PET-CT has been reported to be 61%, 57%, and 87%, respectively. The clinical management of 11% of patients with invasive pancreatic cancer was changed as a result of PET-CT findings. Nevertheless, the role of PET-CT in this setting is evolving and has not yet been established [76,77].

Regarding the technique used for FDG PET-CT for pancreatic cancer staging, the NCCN clearly states that a high-quality, conventional CT with contrast should be performed in addition to a PET-CT, regardless of whether the PET-CT scan is performed with contrast [78]. This is likely because the timing of contrast is critical to the assessment of the relationship of the tumor to the superior mesenteric artery and vein, and the portal, splenic, and superior mesenteric veins to establish surgical candidacy. NCCN guidelines do not include FDG PET-CT for therapeutic assessment or surveillance, although it is widely used clinically for these indications.

Treatment protocols for pancreatic cancer are relatively well established [79]. A common bile duct stent is typically placed early when duct obstruction is present. Curative resection options including Whipple procedure (pancreaticoduodenectomy), +/− pyloric sparing, or total or segmental pancreatectomy. Neoadjuvant chemotherapy +/− radiation may be performed to decrease the tumor size prior to surgery in potentially resectable cases. Advanced or recurrent disease typically relies upon cytotoxic chemotherapy, either gemcitabine monotherapy or a variety of combined chemotherapies [75]. Second line tyrosine kinase inhibitors can be used in resistant cases in tumors expressing NTRK fusions [78].

In general, the magnitude of FDG uptake in primary pancreatic cancer is highly variable, possibly related to the degree of mucinous involvement. However, even largely solid pancreatic cancers can differ in magnitude of metabolic activity (Figure 26) and some highly mucinous tumors can be metabolically active (Figure 27). Evaluating the spread of tumor into the surrounding mesentery is complicated, both on PET and CT, by a host desmoplastic stromal response in pancreatic cancer, which may falsely suggest extra-pancreatic tumor extension. Adjacent nodes may also vary both in size and metabolic activity and are likely better assessed by EUS than either CT or FDG PET-CT. However, even when the primary tumor is highly mucinous and relatively low in uptake, distant metastases are often hypermetabolic (Figure 27c,d). No specific role of FDG PET-CT has been established for evaluation of IPMNs, of which the main duct (not side branch) forms have a high rate of development of carcinoma. However, highly mucinous pancreatic neoplasms may result in pseudomyxoma peritonei if the contents are dispersed or the tumor is broken when it is removed. Pancreatitis, either infectious or autoimmune, can present with increased metabolic activity. In autoimmune pancreatitis (IgGS4 disease), uptake tends to be uniform throughout the entire pancreas (Figure 28). Peritoneal spread of pancreatic cancer is often diffuse and may involve omental caking as well as peritoneal carcinomatosis (Figure 29). Peritonitis can mimic peritoneal carcinomatosis on FDG PET-CT and may complicate pancreatitis (Figure 30). It is safe to assume that the role of FDG PET-CT in pancreatic cancer is evolving. Widespread adoption of FDG PET-CT in clinical practice for the evaluation of pancreatic may precede broad adoption by NCCN guidelines.

## 9. Bowel Adenocarcinoma: Colon, Rectum, Small Bowel

Colorectal cancer is the most common GI malignancy. Although death rates have been dropping due to better screening, it remains the third most common cause of overall cancer deaths. The etiology of colorectal cancer is multifactorial, and includes genetic, dietary factors, and underlying chronic inflammatory conditions such as ulcerative colitis [79]. Genetic factors, including familial adenomatosis polyposis (FAP) and Lynch syndrome, play a role in the development of colorectal cancer, although most cases of colorectal cancer are sporadic. Smoking, older age, obesity and high alcohol intake have also been shown to increase risk of colorectal cancer. Surgery is the definitive treatment, occasionally preceded by neoadjuvant chemotherapy +/− radiation. Advances in cytotoxic, targeting and biological drugs for treatment of advanced colorectal cancer or as an adjunct to surgery in high-risk patients are in rapid development [80]. Colon cancer tends to spread first to adjacent regional and retroperitoneal nodes, but will occasionally spread to the liver without evidence of other disease. Rectal cancer shows a similar pattern of spread but because of the rich vascular plexus around the rectum, early hematogenous spread to the lungs, even in low T- or N-stage disease, may occur. Mucinous signet-ring cell adenocarcinomas tend to spread throughout the peritoneum rather than to the liver [81].

Small-bowel cancer accounts for only 5% of GI tumors, while adenocarcinoma accounts for 40% of small-bowel tumors [82]. The remainder are predominantly neuroendocrine tumors, lymphoma, GIST tumors and metastases (discussed elsewhere). For small-bowel adenocarcinoma, risk factors are similar to those for colon cancer, and include familial syndromes of Lynch syndrome, Peutz–Jeghers syndrome and familial adenomatous polyposis (FAP). Over half of cases occur in the duodenum. Patients tend to be younger than those with colorectal carcinoma [83,84,85]. Small-bowel adenocarcinoma has a propensity for diffuse peritoneal spread, with greater than 1/3 of patients developing peritoneal carcinomatosis [86].

The NCCN imaging guidelines for colon, rectum and small-bowel adenocarcinoma are similar, with minor variations [87,88,89]. For initial staging for bowel adenocarcinoma, NCCN recommends CT of the chest, abdomen and pelvis, +/− MRI. FDG PET-CT is not routinely indicated in the initial workup or staging if metastatic disease is not suspected either clinically or by initial imaging. However, PET-CT is supported if metastatic disease is suspected, if conventional imaging is equivocal on contrast-enhanced CT or MRI, or if the patient cannot receive contrast. In patients with potentially curable M1 disease, such as a single metastasis to the liver that could be resected or treated with liver directed therapy (e.g., ablation or radioembolization), NCCN does endorse the use of PET-CT to identify other sites of metastatic disease that may change the approach. In patients who are candidates for resection of a single metastatic lesion to the liver, 20% will show additional metastases on PET-CT [90]. For monitoring therapeutic response and for surveillance imaging, NCCN guidelines again support the use of PET-CT in assessing the effect of liver directed therapy in Stage IV disease. For small-bowel adenocarcinoma, NCCN supports the use of FDG PET-CT for assessment of peritoneal disease.

NCCN also supports the use of FDG PET-CT in the investigation of serial carcinoembryonic antigen (CEA) elevations when conventional imaging is negative. In post-operative patients with a rising CEA, a pooled sensitivity and specificity of FDG PET-CT of 94.1% and 77.2% has been reported [91]. In post-operative surveillance with an increased CEA, FDG PET-CT is superior to CT alone, but sensitivity of both are proportionally improved as a function of the CEA level [92]. For example, the sensitivity of FDG PET-CT (and CT alone) in identifying recurrent disease is 83.3% (57.1%) with a CEA level of ≤3.4 ng/mL, 96% (78.3%) for a CEA of 3.4–10 ng/mL, 96.7% (86.7%) for a CEA of 10–30 ng/mL, and 100% (93.8%) for a CEA of >30 ng/mL [92]. Despite NCCN recommendations that PET-CT be used only with a negative CT, if performed with a contrast-enhanced CT, FDG PET-CT has been proposed as the first choice in evaluation of post-operative patients with colorectal cancer and a rising CEA [93]. For rectal carcinoma, FDG PET-CT can be considered initially (prior to other imaging modalities) with a serially rising CEA.

The value of FDG PET-CT in adenocarcinoma of the bowel is well established. In general, both the primary tumor and metastatic deposits are hypermetabolic (Figure 31). However, there are several pitfalls and challenges related to PET-CT for colorectal or small-bowel adenocarcinomas. There are false negatives and careful attention to the CT scan is important. Mucinous signet-ring cell variants of adenocarcinoma may be low in metabolic activity, as are their metastases. Mucinous metastases throughout the peritoneum, or pseudomyxoma peritonei, is typically low in activity and attenuation characteristics. Pseudomyxoma peritonei can arise from a variety of both benign and malignant mucinous neoplasms, such as pancreas, colon, ovary and breast, but is most commonly the result of appendiceal mucinous adenocarcinomas (Figure 32) [94]. Tumors of the bowel can also be obscured by physiologic activity within the bowel, or by high bowel uptake due to oral hypoglycemics, such as metformin (Figure 33) [95].

There are numerous potentially false-positive pitfalls and challenges when imaging colorectal or small-bowel adenocarcinoma [96,97]. Inflammatory bowel disease, either infectious or non-infectious, can produce hypermetabolic lesions. These include autoimmune inflammatory bowel disease (Figure 34), neutropenic enterocolitis (typhlitis) (Figure 35) and diverticulitis (Figure 36). Post-radiation inflammatory change may persist for 2–3 months following treatment. In general, therapeutic assessment with FDG PET-CT should be delayed at least 2 months following completion of radiation (Figure 37). Post-treatment fistulas are a significant source of false positives in the tumor treatment bed, and the magnitude of activity can be every bit as high as it is with persistent tumor (Figure 38). A clue to the presence of a fistula is the presence of air in the hypermetabolic tissue, although this may be inapparent. When a fistula is present, the fistulous track and communicating bed will be metabolically active. Low level infection/inflammation is always present. Of course, an abscess can also be intensely hypermetabolic. False-positive foci of activity at anastomoses due to attenuation artifacts are also possible. Non-attenuation corrected PET images may help in this regard, although host-reactive change to the presence of mesh or sutures may also produce uptake. Although there are many choices as to CT quality to be performed with PET-CT, the use of conventional dose parameters and the inclusion of IV and oral contrast help to resolve challenging findings on PET-CT. This is particularly true within the abdomen.

## 10. Anal Carcinoma

Anal cancer arises from the squamous columnar junction of the anus and predominantly due to squamous cell carcinoma. Other types of tumors, such as melanoma and adenocarcinoma, may also involve the anus. Historically accounting for about 1.5% of GI tumors, the incidence of anal carcinoma of the anus is steadily rising primarily due to its increasing frequency in HIV positive males as well as females 50 years of age and older [98]. Overall, anal carcinoma is twice as common in women as in men. Multiple HPV subtypes, chronic immunosuppression, smoking, a history of cervical cancer and Crohn’s disease are other risk factors [99].

Spread of anal cancer is typically first to inguinal lymph nodes, then intrapelvic and retroperitoneal nodes, and finally distant metastases. For localized disease, the five-year survival is favorable, at 81.3%. For advanced disease, prognosis is worse, with a five-year survival of 29.6% [100]. Because resection often results in incontinence and the necessity for creation of a colostomy, definitive initial treatment is usually accomplished by a combined modality chemotherapy and intensity modulated radiation therapy (IMRT) [101]. Metastatic disease at diagnosis is typically treated with a 5FU/cisplatin combination. Options for treatment of refractory metastatic anal cancer include immune check-point inhibitors and other targeted approaches [102,103].

NCCN supports the use of FDG PET-CT in the initial workup of anal carcinoma, although stating that it does not replace diagnostic CT. NCCN also endorses the use of PET-CT in patients with progression or recurrence on digital rectal exam. FDG PET-CT has been shown to detect both the primary tumor as well as involved lymph nodes more often than does CT [104]. PET-CT is particularly effective in identifying small primary tumors and nodal metastases, which complicates the performance of CT. A review and meta-analysis reported that the sensitivity of PET-CT and CT alone was 99% and 60% for detecting the primary tumor, and that PET-CT lead to a change in nodal staging in 28% of patients [105]. Following chemotherapy, they reported that 78% had a complete response by PET. This supports the routine use of FDG PET-CT in patients with anal carcinoma.

Anal carcinoma is typically metabolically active on PET, as are regionally involved nodes and distant metastases (Figure 39). When high anal carcinoma extends into the lower rectum, the pattern of spread may be more in keeping with that of rectal cancer than anal carcinoma, with intrapelvic nodes and lung metastases (Figure 40). Perianal infections (Figure 41), inflammatory bowel disease, inflamed hemorrhoids (Figure 42) and even normal anal sphincter muscles (Figure 43) may be confused with pathologic processes [106]. However, FDG PET-CT remains an important imaging modality in the management of patients with anal carcinoma.

## 11. Conclusions

The role of FDG PET-CT is well established in many of the most common gastrointestinal (GI) tumors for staging, therapy assessment and surveillance. Despite this, there are many pitfalls and nuances that create challenges in interpreting FDG PET-CT scans in patients with GI tumors. An awareness of these limitations and challenges is essential, because a misleading or misinterpreted study can have untoward therapeutic consequences for the patient. An awareness of the patient history is particularly critical in accurate interpretation of FDG PET-CT. A high value must be placed on concurrent and longitudinal correlative imaging studies (CT and MRI). To this end, a high-quality CT scan with oral and appropriately timed intravenous contrast has significant value when performed as a component of the PET-CT scan. Close communication between the imaging professionals and the referring providers adds value to the quality and relevance of the imaging report. In addition, the referring providers should understand the limitations and advantages of PET-CT, communicate well with the imaging professionals, and ask specific questions that will guide the imagers in providing the most relevant and accurate information.

## Figures and Tables

**Figure 1 cancers-14-02668-f001:**
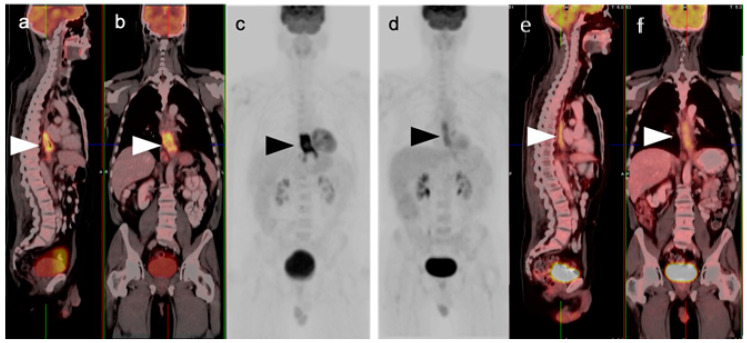
Pre- and post-treatment esophageal cancer. (**a**–**f**) FDG PET-CT pre-treatment for esophageal cancer. Sagittal (**a**), coronal (**b**) and anterior maximum intensity projection (MIP) (**c**) images show a hypermetabolic distal esophageal carcinoma (arrowheads); (**d**–**f**) Post-treatment esophageal cancer in the same planes as the pre-treatment scan. Repeat FDG PET-CT 3 weeks following completion of neoadjuvant chemoradiation show mild persistent metabolic activity through the radiation field (arrowheads). Imaging at 10–12 weeks post-completion of chemoradiation is recommended to minimize false FDG PET-CT findings such as these, due to post-treatment inflammatory changes.

**Figure 2 cancers-14-02668-f002:**
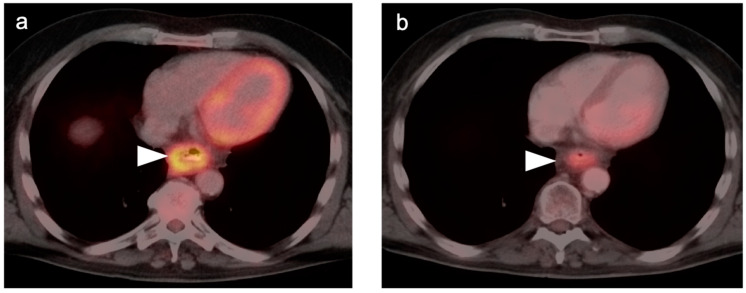
Persistent tumor following CRT. (**a**) Pre-treatment axial fused FDG PET-CT demonstrates a hypermetabolic tumor in the mid-esophagus (white arrowhead); (**b**) Axial fused FDG PET-CT image 12 weeks following completion of CRT demonstrates persistent, although improved, hypermetabolic soft tissue mass in the mid-esophagus (white arrowhead). Biopsy confirmed the presence of persistent viable tumor.

**Figure 3 cancers-14-02668-f003:**
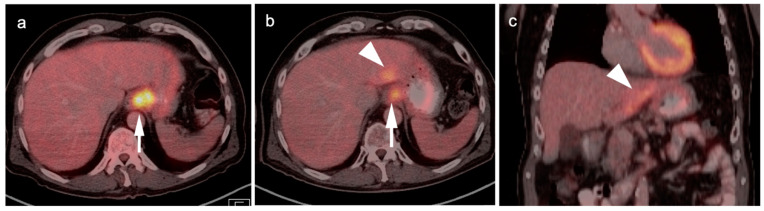
Esophageal cancer pre- and post-treatment. (**a**) Pre-treatment axial fused FDG PET-CT image shows a circumferential hypermetabolic esophageal tumor (white arrow); (**b**,**c**) Axial (**b**) and coronal fused FDG PET-CT images five weeks post-neoadjuvant CRT demonstrate residual soft tissue thickening and metabolic activity (white arrow), although improved from pre-treatment. This would require biopsy to distinguish residual viable tumor from post-treatment inflammatory change. In addition, there is increased metabolic activity along the left border of the liver (white arrowhead) likely due to the liver being within the radiation field.

**Figure 4 cancers-14-02668-f004:**
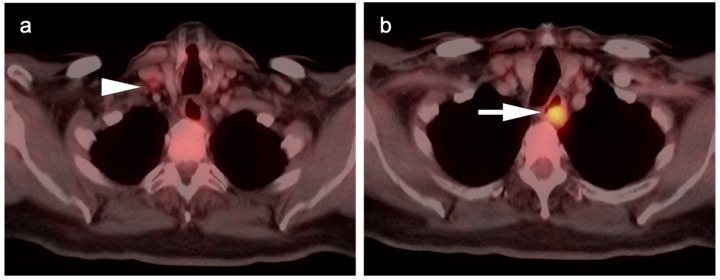
Upper thoracic esophageal cancer. (**a**) Axial fused FDG PET-CT axial image demonstrates a single hypermetabolic low right level 3 cervical node (white arrowhead); (**b**) Axial fused FDG PET-CT image shows that the primary tumor is a hypermetabolic eccentric mass in the upper thoracic esophagus (white arrow). Upper esophageal tumors frequently have nodal spread to the low neck or supraclavicular region.

**Figure 5 cancers-14-02668-f005:**
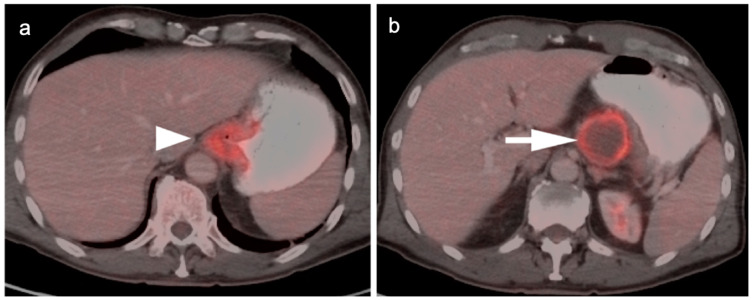
FDG PET-CT images of the upper abdomen in a patient with a GE junction esophageal cancer, (**a**) Axial fused FDG PET-CT show a GE junction tumor extending slightly into the gastric cardia (white arrowhead); (**b**) Axial fused FDG PET-CT image shows a typical pattern of nodal spread to lymph nodes in the gastrohepatic ligament (white arrow).

**Figure 6 cancers-14-02668-f006:**
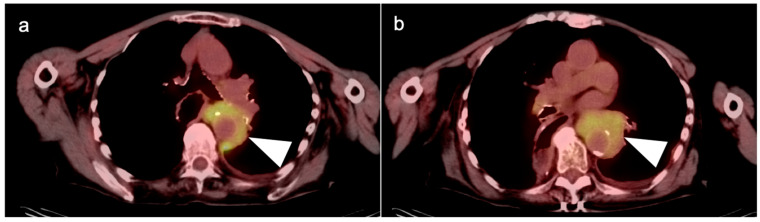
(**a**,**b**) Axial fused FDG PET-CT images of the chest show a recurrent mid-esophageal cancer following CRT and esophageal resection and gastric pull-up. The hypermetabolic tumor on FDG PET-CT is infiltrative and wraps around the aorta (white arrowheads).

**Figure 7 cancers-14-02668-f007:**
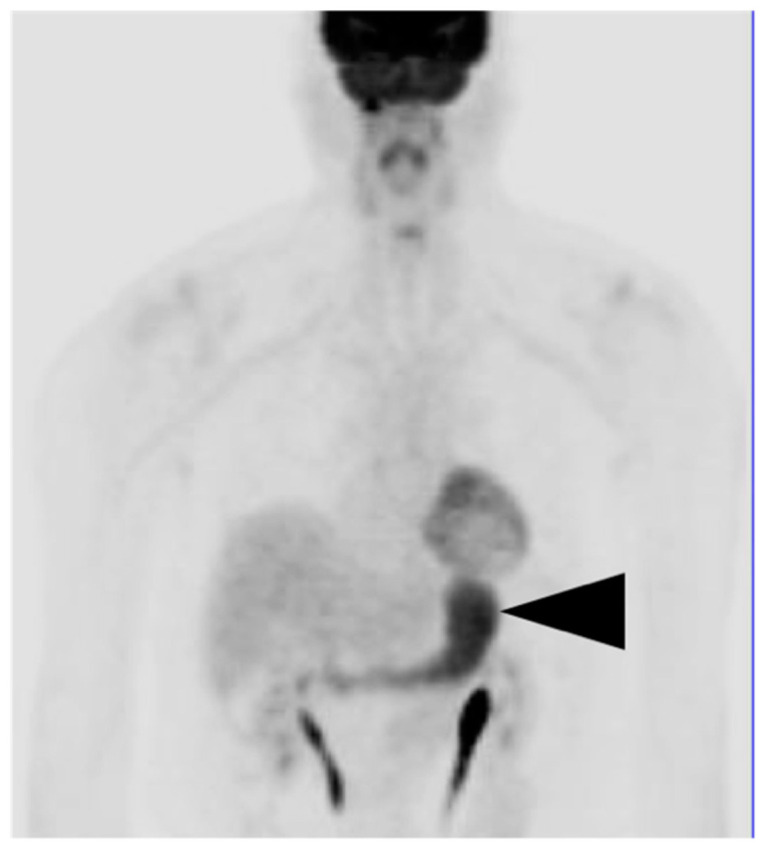
Physiological gastric uptake of FDG. Anterior FDG PET MIP image shows diffuse metabolic activity throughout the stomach (black arrowhead), This is nonspecific and may be related to gastritis or physiological or unknown factors. Tumor cannot be excluded. If this is an incidental finding, endoscopy is recommended if the patient has gastric symptoms.

**Figure 8 cancers-14-02668-f008:**
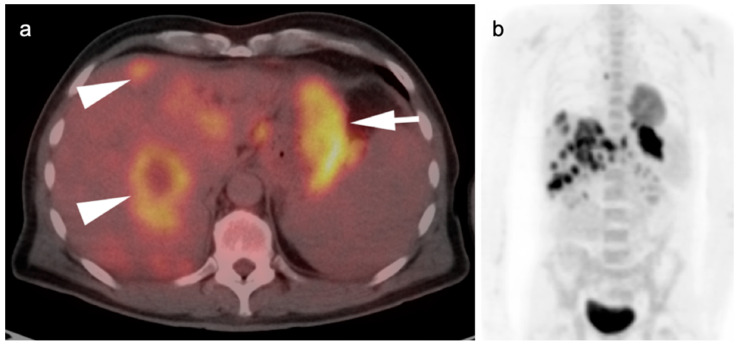
Metastatic gastric cancer. (**a**) Axial FDG PET-CT and (**b**) MIP image. Markedly hypermetabolic in gastric carcinoma (white arrow) with widespread hepatic metastases (white arrowheads).

**Figure 9 cancers-14-02668-f009:**
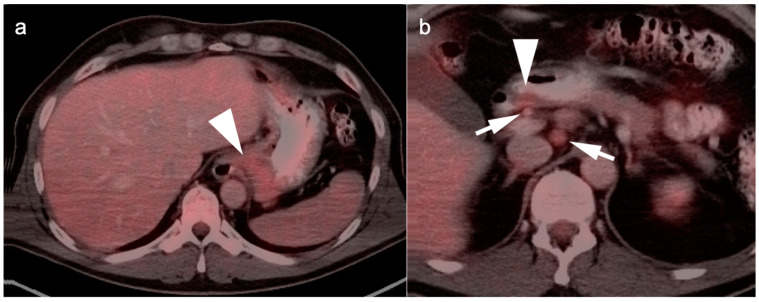
Two cases of gastric carcinoma on FDG PET-CT. (**a**) Axial fused FDG PET-CT image of the upper abdomen shows mild metabolic activity in the gastric cardia (with tumor involvement, white arrow). This degree of metabolic activity is similar to that in many normal patients; (**b**) Magnified axial fused FDG PET-CT image. A small nodular tumor on the posterior wall of the gastric antrum is only mildly metabolically active (white arrowhead) but is associated with two small hypermetabolic lymph nodes (white arrows) consistent with regional nodal spread.

**Figure 10 cancers-14-02668-f010:**
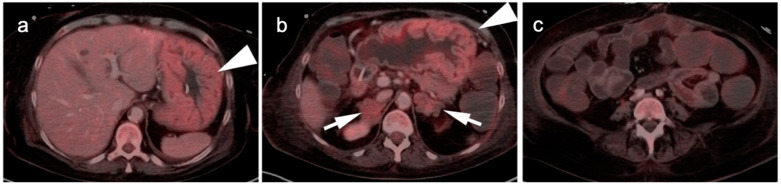
Recurrent gastrinoma. Fused axial FDG PET_CT images of the abdomen show hypertrophy of the parietal cells in the gastric fundus because of elevated gastrin production. This can mimic gastric cancer as is shown in this case ((**a**,**b**), white arrowheads). Bilateral adrenal metastases are present ((**b**), white arrows) There are also dilated fluid-filled loops of bowel due to a hypersecretory state (**c**).

**Figure 11 cancers-14-02668-f011:**
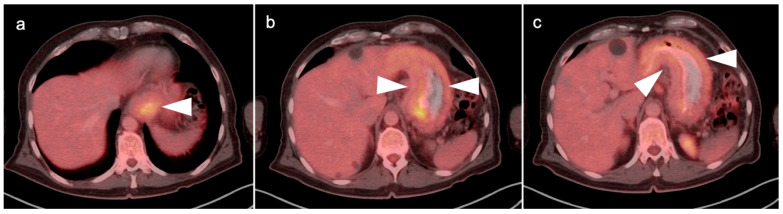
Linitus plastica. (**a**–**c**) Diffusely thickened, featureless “leather bottle” gastric configuration, mildly hypermetabolic, as shown on axial fused FDG PET-CT images due to diffuse infiltrative adenocarcinoma of the stomach (white arrowhead).

**Figure 12 cancers-14-02668-f012:**
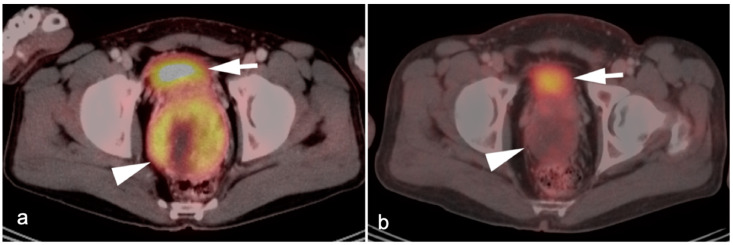
A large rectal carcinoid tumor before (**a**) and after (**b**) treatment with Imatinib. Fused axial PET-CT images of the low pelvis show a significant decrease in metabolic activity with treatment. Urinary bladder is shown by white arrow.

**Figure 13 cancers-14-02668-f013:**
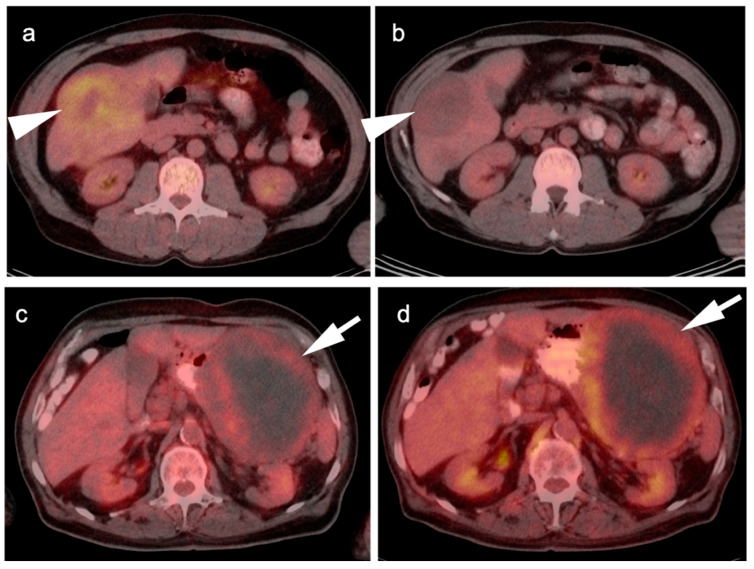
Two cases of GIST tumor. Case 1 (**a**,**b**): Hepatic GIST metastatic disease before and after successful treatment with Imatinib. (**a**) Pretreatment axial FDG PET-CT shows hypermetabolic mass in the right lobe of the liver (white arrowhead); (**b**) Post treatment axial fused FDG PET-CT image show resolution of metabolic activity with treatment (white arrowhead). Complete metabolic response means disease control, not complete pathologic response; Case 2 (**c**,**d**): Large gastric GIST tumor before and after treatment with imatinib. (**a**) Pretreatment axial FDG PET-CT shows a large, centraly necrotic, peripherally hypermetabolic tumor arising from the stomach (white arrow); (**b**) Post-treatment axial FDG PET-CT shows that the tumor has increased both in size and metabolic activity (white arrow), consistent with lack of tumor control.

**Figure 14 cancers-14-02668-f014:**
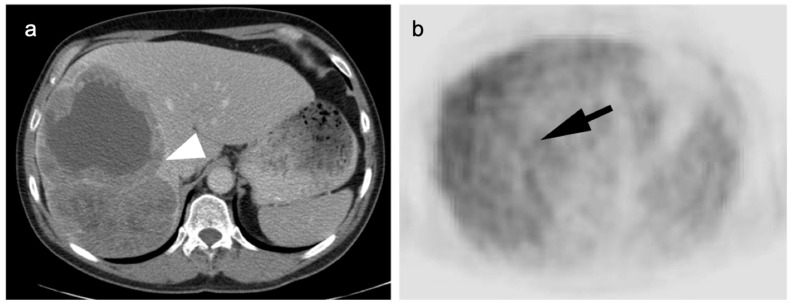
Well-differentiated HCC shown on contrast-enhanced CT (**a**) and axial FDG PET (**b**). The large lesion on CT (white arrowhead) is isometabolic to the remainder of the liver on FDG PET-CT (black arrow), which is a typical feature of well-differentiated HCC.

**Figure 15 cancers-14-02668-f015:**
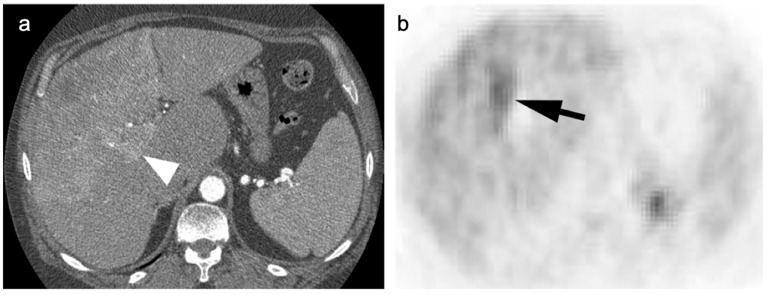
(**a**) A biopsy-proven well-differentiated HCC shows diffuse arterial enhancement on contrast-enhanced CT (white arrowhead); (**b**) On axial FDG PET, this mass is largely isometabolic to the liver, although a central more focally hypermetabolic regions may indicate a region of more poorly differentiated tumor (black arrow).

**Figure 16 cancers-14-02668-f016:**
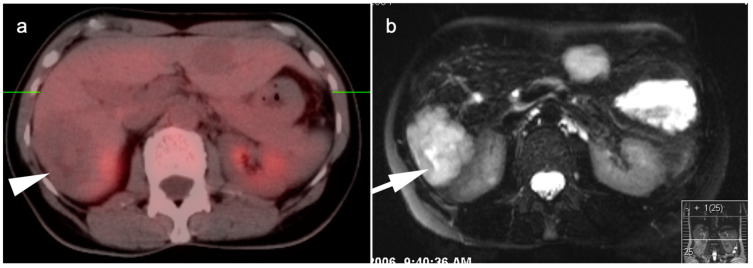
(**a**) A large cavernous hemangioma of the right lobe of the liver is isometabolic to the normal liver on axial fused FDG PET-CT; (**b**) On T2 axial MRI, this lesion shows typical bright signal seen in cavernous hemangiomas.

**Figure 17 cancers-14-02668-f017:**
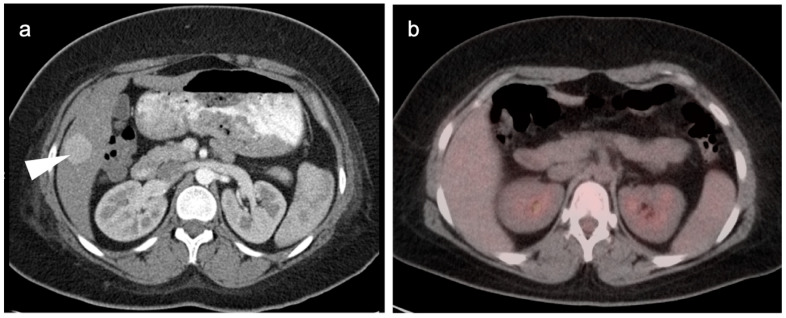
(**a**) A small enhancing nodule in the right lobe of the liver on axial contrast-enhanced CT was biopsy-proven to be an adenoma; (**b**) On a fused axial FDG PET-CT this lesion is isometabolic to the liver.

**Figure 18 cancers-14-02668-f018:**
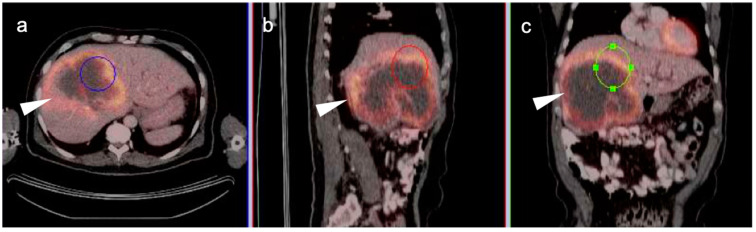
Intrahepatic cholangiocarcinoma. Axial (**a**), sagittal (**b**) and coronal (**c**) projections of a fused FDG PET-CT scan show a typical hypermetabolic ring-shaped area of increased metabolic activity with central necrosis (white arrowheads). SUVmax in shown region of interest (ROI) is 10.3.

**Figure 19 cancers-14-02668-f019:**
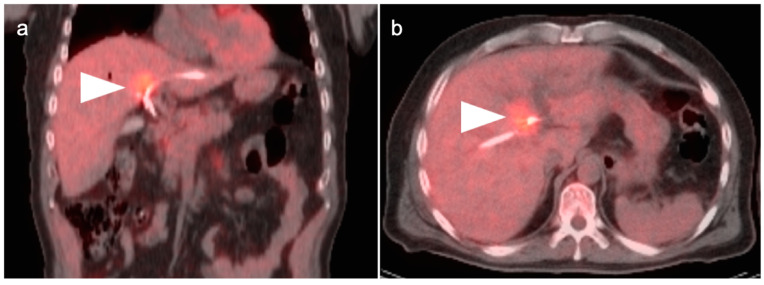
Hilar cholangiocarcinoma (Klatskin tumor). The tumor at the hepatic hilum is moderately hypermetabolic (white arrowheads) on fused FDG PET-CT shown in coronal (**a**) and axial (**b**) projections, which required placement of biliary drainage tubes.

**Figure 20 cancers-14-02668-f020:**
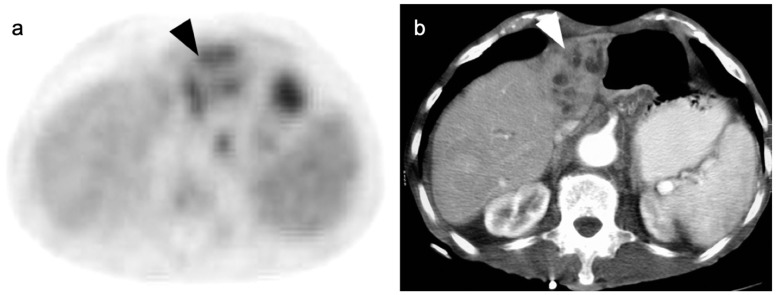
A small cholangiocarcinoma resulted in blockage of the bile ducts draining a portion of the left lobe of the liver. (**a**) Axial FDG PET of the upper abdomen shows hypermetabolic bile ducts (black arrowhead); (**b**) The bile ducts are dilated on axial contrast-enhanced CT (white arrowhead). The findings are consistent with post-obstructive focal cholangitis. This can result in a false-positive FDG PET-CT scan performed for cancer assessment.

**Figure 21 cancers-14-02668-f021:**
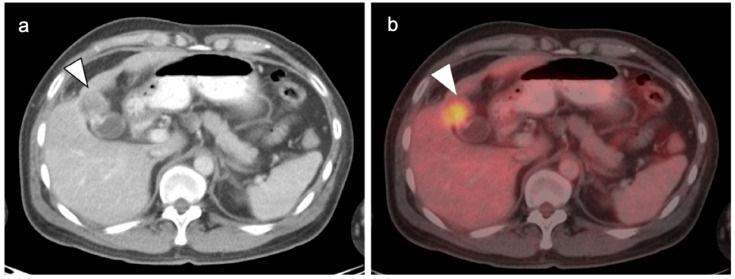
(**a**) Polypoid mass in the fundus of the gallbladder on axial contrast-enhanced CT was a gallbladder carcinoma at surgery (white arrowhead); (**b**) Concurrent fused axial FDG PET-CT image shows focal metabolic activity in the tumor (white arrowhead).

**Figure 22 cancers-14-02668-f022:**
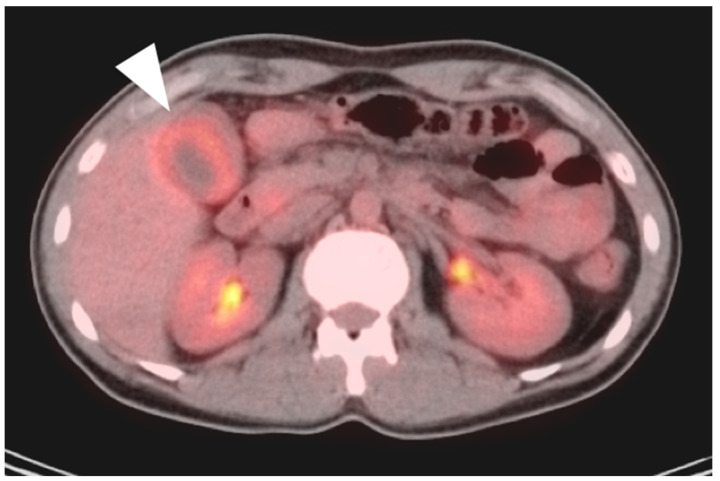
An axial fused FDG PET-CT image shows gallbladder wall thickening and diffuse increased metabolic activity within the liver consistent with cholecystitis, either infectious or inflammatory (white arrowhead). There is slight mis-registration between the CT and PET due to differences in breathing between the two exams.

**Figure 23 cancers-14-02668-f023:**
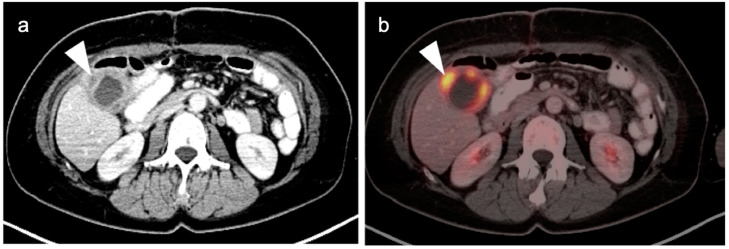
Xanthogranulomatous cholecystitis. (**a**) Contrast-enhanced CT demonstrates gallbladder wall thickening with nodular hypoattenuating regions within the gallbladder wall (white arrowhead); (**b**) Axial FDG PET-CT shows focal increased metabolic activity in the hypoattenuating nodular regions within the gallbladder wall, due to accumulation of lipid laden macrophages in areas of chronic inflammation (white arrowhead).

**Figure 24 cancers-14-02668-f024:**
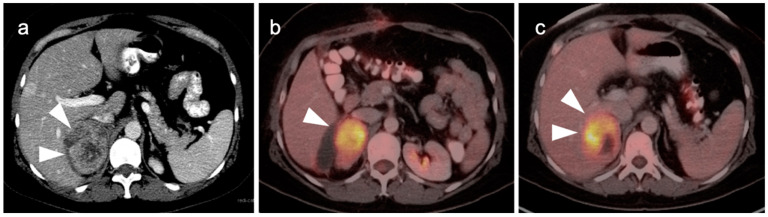
Adrenal cortical carcinoma. (**a**) A preoperative axial contrast-enhanced CT shows a heterogeneously enhancing right adrenal mass with areas of necrosis which was an adrenal cortical carcinoma at pathology; (**b**,**c**) Axial fused images from an FDG PET-CT scan performed 7 months following resection of the adrenal cortical carcinoma showing a recurrent hypermetabolic mass in the right adrenal fossa ((**b**,**c**), white arrowheads), with the suggestion of possible invasion into the adjacent liver, with an indistinct plane between the mass and the liver (white arrowheads, (**c**)).

**Figure 25 cancers-14-02668-f025:**
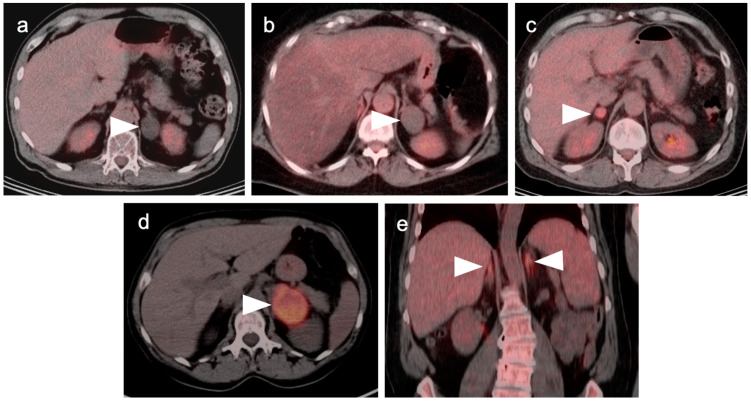
Variable appearance of adrenal lesions (white arrowheads) on FDG PET-CT: (**a**) Axial fused FDG PET-CT. A left adrenal nodule (white arrowhead) is low in attenuation (<10 HU) and is without appreciable metabolic activity, typical findings of lipid-rich adenoma; (**b**) Axial fused FDG PET-CT. Attenuation is slightly higher (HU 15) and metabolic activity is lower in this adrenal nodule, compared to the metabolic activity of the liver, supporting that this is likely a benign nodule but warrants follow-up for confirmation of stability. Note there is also geographic hepatic steatosis on CT; (**c**) Axial fused FDG PET-CT. A small adrenal nodule demonstrates metabolic activity 1.8 times that in the liver. This is suspicious for malignancy and further workup is warranted; (**d**) Axial fused FDG PET-CT. A large intensely hypermetabolic adrenal mass is a metastasis vs. adrenal cortical cancer (in this case it was a metastasis from a primary was non-small cell lung cancer); (**e**) Coronal fused FDG PET-CT image. Symmetrical hypermetabolism in both adrenal glands without nodularity may represent stressed-induced adrenal activation or adrenal hyperplasia.

**Figure 26 cancers-14-02668-f026:**
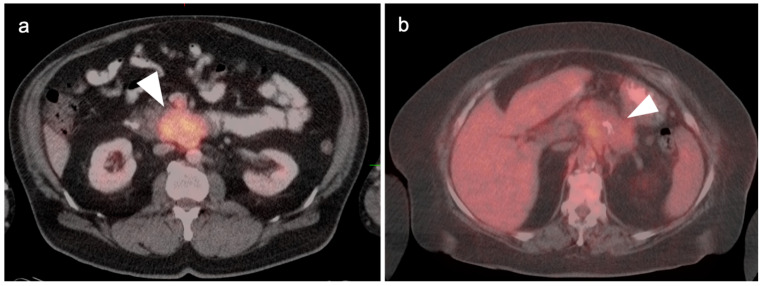
Variable appearance of pancreatic cancer on axial fused FDG PET-CT images in two patients. Case 1: (**a**) Intensely hypermetabolic primary tumor involving the uncinate process of the pancreas (white arrowhead); Case 2: (**b**) Mildly hypermetabolic tumor involving the body of the pancreas (white arrowhead). Both tumors are solid in appearance on CT, and not grossly cystic, and both were adenocarcinoma.

**Figure 27 cancers-14-02668-f027:**
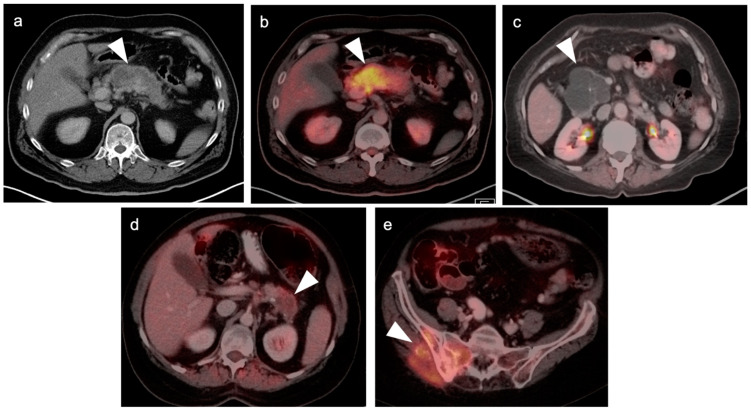
Highly variable appearance of highly mucinous pancreatic neoplasms on FDG PET-CT: (**a**–**e**). (**a**,**b**) Mucinous adenocarcinoma of the body of the pancreas is low attenuation on axial contrast-enhanced CT (**a**) and yet highly metabolically active on axial fused FDG PET-CT (**b**) (white arrowheads); (**c**) Complex cystic mass in the head of the pancreas is a highly mucinous cystadenocarcinoma and shows no metabolic activity on axial fused FDG PET-CT (white arrowhead). The differential diagnosis would include pseudocyst from prior pancreatitis; (**d**) A mucinous adenocarcinoma of the tail of the pancreas shows only mild uptake on axial fused FDG PET-CT (white arrowhead); (**e**) The same pancreatic mass as (**d**) has metastasized to the right iliac bone and sacrum (white arrowhead). Although the pancreatic mass shows only mild metabolic activity, the pelvic metastasis from this same tumor is strongly hypermetabolic.

**Figure 28 cancers-14-02668-f028:**
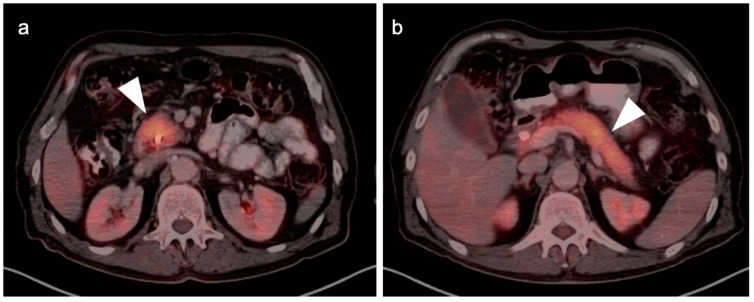
IgGS4-associated autoimmune pancreatitis in a 72-year-old with biliary obstruction and abdominal pain. Axial fused FDG PET-CT images of the head (**a**) and body/tail (**b**) of the pancreas (white arrowheads) show that the pancreas is diffusely hypermetabolic, somewhat featureless, sausage-shaped and larger than normal for an older person.

**Figure 29 cancers-14-02668-f029:**
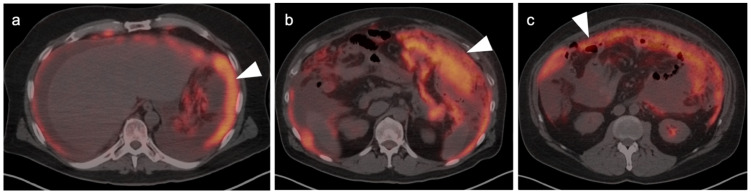
Peritoneal carcinomatosis from recurrent pancreatic cancer. Axial fused FDG PET-CT images. (**a**) The peritoneal surface is diffusely lined with tumor (white arrowhead). (**b**,**c**) There is omental caking with hypermetabolic tumor (white arrowheads).

**Figure 30 cancers-14-02668-f030:**
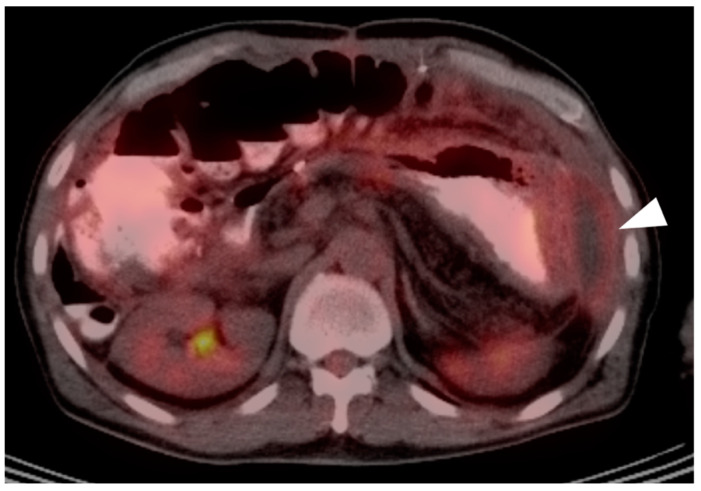
Bacterial and chemical peritonitis resulting from pancreatitis. On an axial fused PET-CT image, diffuse metabolically active tissue is present throughout the peritoneal space with fluid collections, consistent with abscesses (white arrowhead). This can mimic peritoneal carcinomatosis.

**Figure 31 cancers-14-02668-f031:**
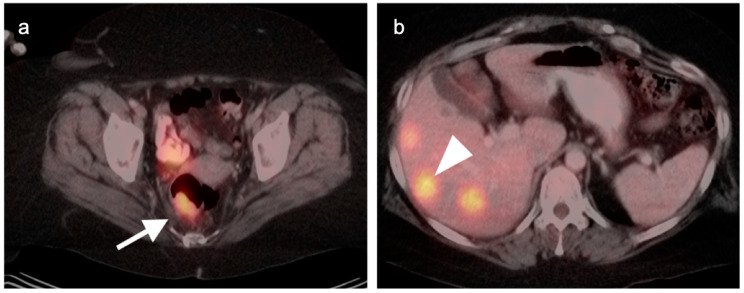
Rectal carcinoma with hepatic metastases, (**a**) Axial fused PET-CT image of the pelvis shows a hypermetabolic rectal mass (white arrow); (**b**) Axial fused PET-CT image of the upper abdomen shows multiple hypermetabolic liver metastases (white arrowhead). Typically, both primary colorectal carcinomas and their metastases are intensely hypermetabolic, although mucinous tumors can be low in activity.

**Figure 32 cancers-14-02668-f032:**
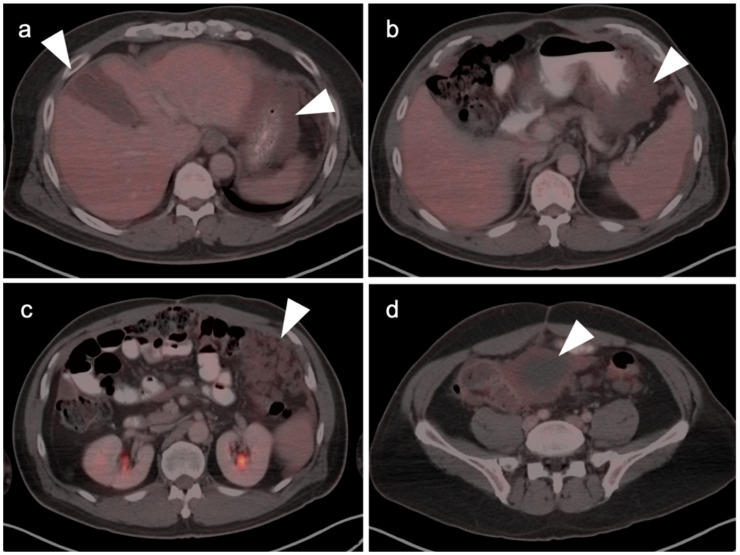
Pseudomyxoma peritonei shown on multiple axial fused FDG PET-CT images. Spillage or seeding of mucinous material from either mucinous cystadenomas or cystadenocarcinomas of the pancreas (as well as breast, ovary, colon and appendix) can result in gelatinous ascites (white arrowheads) that will spread and grow throughout the abdomen. Shows is a case with (**a**) Mucinous material in the gallbladder fossa and implanted on the stomach (**a**,**b**); (**c**) Infiltrating mucinous implants caking the omentum; and (**d**) Mass-like mesenteric implants.

**Figure 33 cancers-14-02668-f033:**
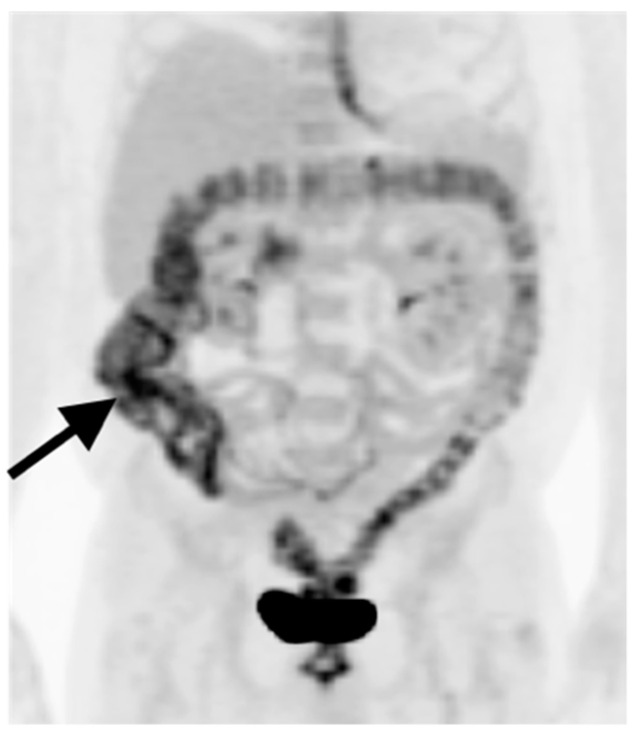
Metformin effect. Anterior MIP FDG PET image of the abdomen. Oral hypoglycemics such as metformin act in part by excreting glucose into the gut (black arrow). This can result in diffuse gut activity (primarily colonic) which can obscure small colorectal sites of tumor involvement.

**Figure 34 cancers-14-02668-f034:**
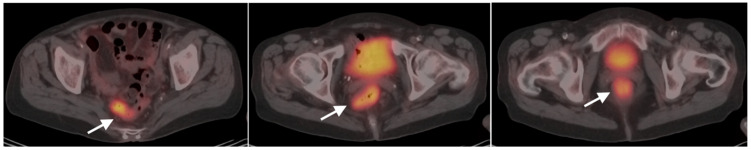
Ulcerative colitis. Axial fused PET-CT images of the pelvis. Hypermetabolic thickened and featureless appearance of the bowel wall extending to the rectum (white arrows) are typical for active ulcerative colitis.

**Figure 35 cancers-14-02668-f035:**
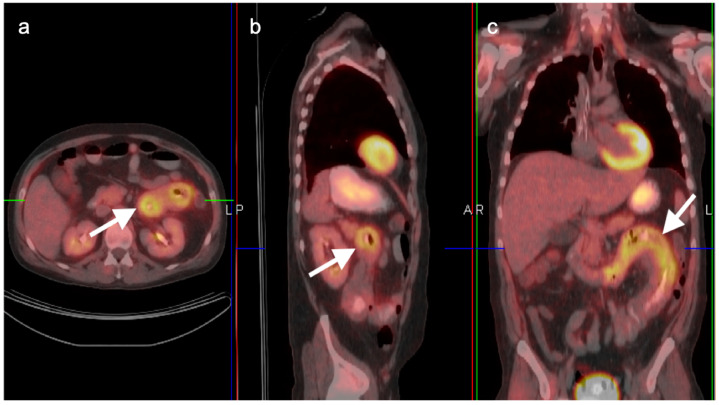
Typhlitis. Fused FDG PET-CT images in axial (**a**), sagittal (**b**) and coronal (**c**) planes. In this neutropenic patient, segmental bowel wall thickening and increased metabolic activity in the distal duodenum/proximal jejunum (white arrows) is typical in appearance for typhlitis, although a more common location is terminal ileum.

**Figure 36 cancers-14-02668-f036:**
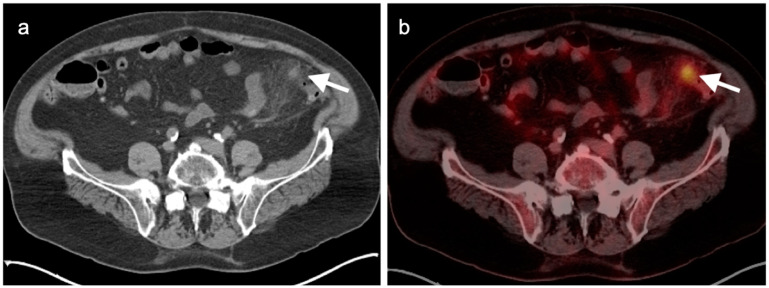
Diverticulitis. Typical features of diverticulitis are shown on axial CT (**a**) and corresponding axial fused FDG PET-CT (**b**) of the lower abdomen. There is an intensely hypermetabolic nodule (white arrow) surrounded by mildly hypermetabolic soft tissue stranding. Differentiation from colon cancer can be difficult in some cases and colonoscopy is typically recommended following treatment for presumptive diverticulitis.

**Figure 37 cancers-14-02668-f037:**
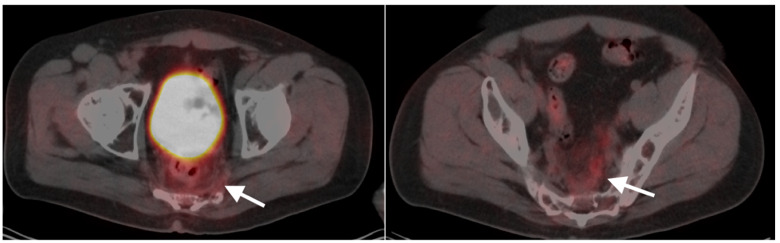
Post-radiation inflammatory change. Axial fused FDG PET-CT images of the pelvis. Presacral thickening/stranding (white arrows) that is metabolically active is due to inflammation secondary to radiation. These findings can persist for up to 2 months or more.

**Figure 38 cancers-14-02668-f038:**
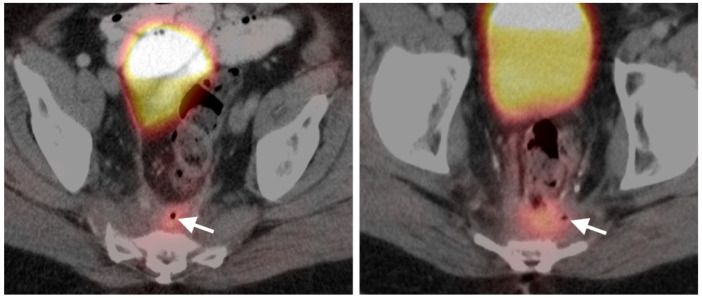
Fistula. Axial fused FDG PET-CT images of the pelvis. Increased metabolic activity on FDG PET-CT in an area of soft tissue thickening and the presence of a small focus of air (white arrows) are clues as to the presence of a post-treatment fistula. These are chronically inflamed and can also be associated with chronic abscesses.

**Figure 39 cancers-14-02668-f039:**
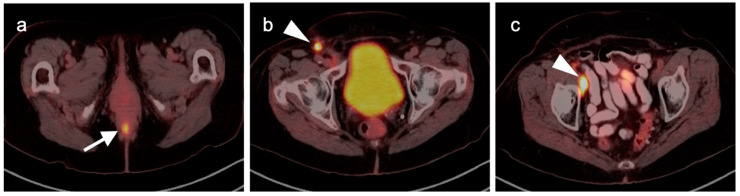
Anal cancer. Axial fused FDG PET-CT images of the pelvis. (**a**) Small primary tumor just within the anus (white arrow); (**b**) Nodal spread of tumor to a small but intensely hypermetabolic right inguinal node, (white arrowhead); (**c**) A right distal external iliac node (white arrowhead).

**Figure 40 cancers-14-02668-f040:**
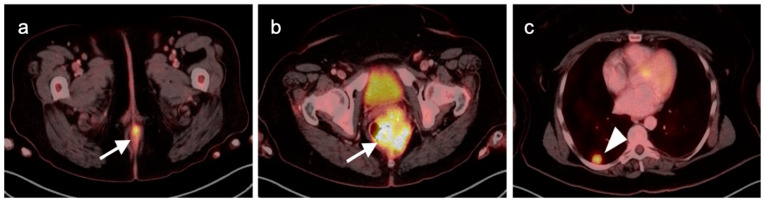
Large anorectal cancer. (**a**) Squamous cell carcinoma of the anus (white arrow), hypermetabolic on an axial fused FDG PET-CT of the low pelvis; (**b**) On an axial fused FDG PET-CT image of the pelvis, tumor extends into the rectum as a bulky hypermabolic mass (white arrow); (**c**) In keeping with behavior of rectal tumors, anal cancer that extends significantly into the rectum is frequently associated with lung metastases, as in this case on an axial fused PET-CT of the low chest (white arrowhead).

**Figure 41 cancers-14-02668-f041:**
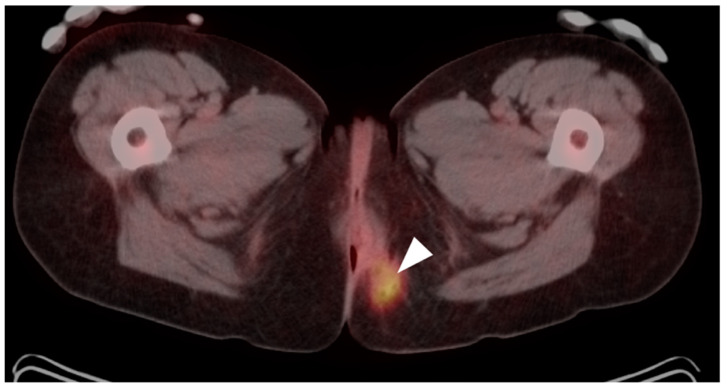
Perianal infection (white arrowhead) is hypermetabolic and can mimic the appearance of an anal cancer, as shown on this axial fused FDG PET-CT of the perineum.

**Figure 42 cancers-14-02668-f042:**
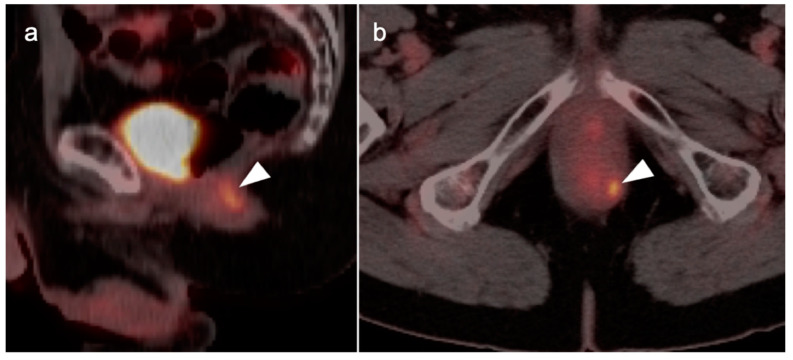
Inflamed internal hemorrhoid (white arrowhead) is hypermetabolic, and may assume a short linear configuration, as shown in this case on sagittal (**a**) and axial (**b**) fused FDG PET-CT images of the pelvis. Inflamed internal or external hemorrhoids can mimic anal cancer.

**Figure 43 cancers-14-02668-f043:**
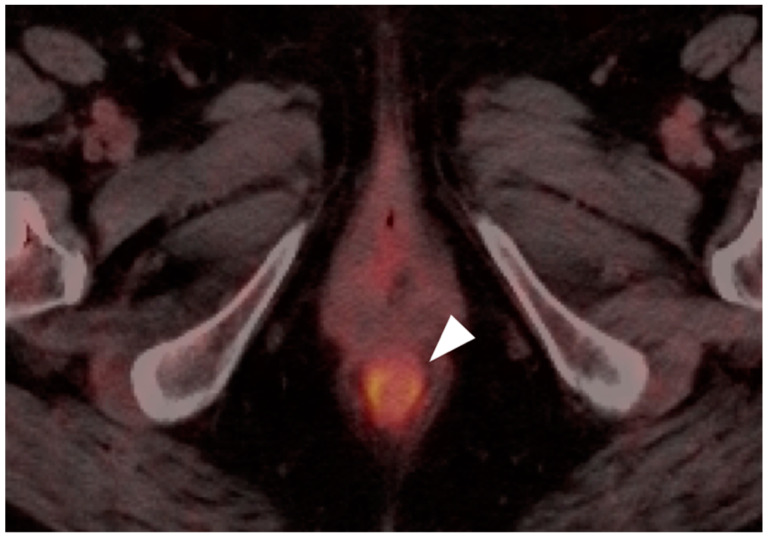
Normal anal sphincter (white arrowhead) is often hypermetabolic, typically assumes a circular shape (as in this axial fused FDG PET-CT of the low pelvis) and can mimic anal cancer.

**Table 1 cancers-14-02668-t001:** Comparison of response criteria for GISTs.

RECIST Criteria	Response	Choi Criteria
Disappearance of all lesions. No new lesions.	Complete (CR)	Disappearance of all lesions. No new lesions.
30% decrease. No further increase.	Partial (PR)	≥10% decrease in size OR ≥15% decrease in density (HU on CT). No new lesions. No obvious progression of non-measurable disease.
Does not meet criteria for PR or PD	Stable disease (SD)	Does not meet criteria for complete response, partial response of progression. No clinical deterioration attributed to tumor progression
20% increase in size AND criteria for CR, PR or SD not met before increased disease.	Progression (SD)	≥10% increase in tumor size AND does not meet criteria of PR by tumor density on CT. New lesions. New intratumoral nodules or increase in size of previous intratumoral nodules.

## Data Availability

There is no data reported.

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
