# Peer review of "PET-CT in Clinical Adult Oncology: III. Gastrointestinal Malignancies"

_cancers, 2022, doi:10.3390/cancers14112668_

Round 1
Reviewer 1 Report
This manuscirpt could be accepted
Author Response
Reviewer 1 had no suggestions for changes. Thank you for efforts and your positive comments.
Reviewer 2 Report
A useful review
Author Response
Reviewer 2 had no suggestions for changes. Thank you for your efforts and positive comments.
Reviewer 3 Report
This peer-reviewed article presents a wide variety of examples of FDG PET-CT applications for staging, therapy, and surveillance in cases of the most common gastrointestinal cancers. Although the technique is well established, the authors point out many pitfalls and nuances that can be challenging to interpret and should therefore be of particular concern to all those interpreting FDG PET-CT scans.
This comprehensive review is a good resource for all professionals in the field, but a few minor issues need to be addressed before publication.
Minor comments:
- Some minor language and spelling corrections should be done. Like line 67, 73, 88, 347 etc.
- For all the figures presented, the plane of the scans presented is not given, which is not a problem for specialists in the field, but Cancers has a wider readership, and this article may be of broader interest.
- Figure 27 b. One of the arrowheads points to the thoracic vertebral body. What does this have to do with adenocarcinoma of the body of the pancreas?
If I were to make specific comments about the weaknesses of this peer-reviewed manuscript, it would be difficult for me to do so because the authors have clearly informed the reader at the beginning of the manuscript what it is and what it is not. Therefore, I found no major flaws or weaknesses-this is a well-written, valuable, and very interesting conglomeration of the latest knowledge on PET-CT in adult clinical oncology. I only noted minor technical problems that should be corrected before publication - which I personally recommend.
If I had to list the strengths of this article, I would say that the authors focus on the pitfalls, nuances, and details of FDG PET-CT that should be considered every time patients are diagnosed. They provide detailed notes on each of the classes of tumors described in the article, in the following (well separated in subsections) organs of the gastrointestinal tract. And this is the strongest point of this manuscript - alerting readers (oncologists) to specific nuances in specific tumor cases when evaluating the results and even the use of FDG PET-CT, based on specific literature examples. In addition, the comparative visualization is an additional strength of this article.
Author Response
Reviewer 3 Comments (responses are in red)
This peer-reviewed article presents a wide variety of examples of FDG PET-CT applications for staging, therapy, and surveillance in cases of the most common gastrointestinal cancers. Although the technique is well established, the authors point out many pitfalls and nuances that can be challenging to interpret and should therefore be of particular concern to all those interpreting FDG PET-CT scans.
This comprehensive review is a good resource for all professionals in the field, but a few minor issues need to be addressed before publication.
Minor comments:
1. Some minor language and spelling corrections should be done. Like line 67, 73, 88, 347 etc.
We have re-read the manuscript carefully and hopefully have corrected the language and spelling errors.
2. For all the figures presented, the plane of the scans presented is not given, which is not a problem for specialists in the field, but Cancers has a wider readership, and this article may be of broader interest.
We have addressed every figure and have carefully labeled them.
3. Figure 27 b. One of the arrowheads points to the thoracic vertebral body. What does this have to do with adenocarcinoma of the body of the pancreas?
We have removed the misplaced arrow from the figure. Thank you for noticing this.
If I were to make specific comments about the weaknesses of this peer-reviewed manuscript, it would be difficult for me to do so because the authors have clearly informed the reader at the beginning of the manuscript what it is and what it is not. Therefore, I found no major flaws or weaknesses-this is a well-written, valuable, and very interesting conglomeration of the latest knowledge on PET-CT in adult clinical oncology. I only noted minor technical problems that should be corrected before publication - which I personally recommend.
If I had to list the strengths of this article, I would say that the authors focus on the pitfalls, nuances, and details of FDG PET-CT that should be considered every time patients are diagnosed. They provide detailed notes on each of the classes of tumors described in the article, in the following (well separated in subsections) organs of the gastrointestinal tract. And this is the strongest point of this manuscript - alerting readers (oncologists) to specific nuances in specific tumor cases when evaluating the results and even the use of FDG PET-CT, based on specific literature examples. In addition, the comparative visualization is an additional strength of this article.